# Resistome phylodynamics of multidrug-resistant *Shigella* isolated from diarrheal patients

Asaduzzaman Asad,[1] Md. Abu Jaher Nayeem,[1] Md. Golam Mostafa,[1] Ruma Begum,[1] Shah Nayeem Faruque,[1] Suraia Nusrin,[2] Israt Jahan,[1] Shoma Hayat,[1] Zhahirul Islam[1]

**ABSTRACT** Multi-drug resistance (MDR) in *Shigella* continues to pose a significant public health challenge, particularly in developing countries. Recent advances in genomics strengthen the surveillance of MDR-pathogens and antimicrobial resistance (AMR) mediators. However, genome-based investigations into resistome dynamics in *Shigella* are limited, specifically in Bangladesh. Therefore, we investigated MDR-*Shigella* resistomes to evaluate their AMR transmission and phylodynamics. Clinical *Shigella* strains were screened for MDR phenotypes through susceptibility tests against 28 antibiotics from 10 different classes. Whole-genome sequencing (WGS) and bioinformatics approaches were performed to unveil the resistome dynamics: >500 global plasmid entities and >1,000 plasmid-mediated resistance gene clusters from global databases were included in this study. We identified 28 distinct antimicrobial resistance genes (ARGs) from nine antibiotic classes, with 75% originating from plasmids. Notably, two conjugative MDR plasmids included nearly all potential ARGs, conferring resistance to first-line drugs for shigellosis. Two third-generation cephalosporin-resistant [*wubC-bla*$_{CTX-M-15}$-ISEcp*1* and *bla*$_{TEM-1}$] and two macrolide-resistant mobile genomic islands (GIs) [*mphA-mrx-mph(R)A*-IS*6100* and *mphE-msrE*-IS*482*-IS*6*] had emerged in *Shigella* in Bangladesh. In addition, trimethoprim-aminoglycoside-streptothricin-sulfonamide-resistant *dfrA1-sat1-aadA1* and *aph3-dfrA14-aph6-sul2* were in conjugative plasmids in Bangladesh. The MDR plasmids and resistant GIs were phylogenetically relevant to Europe, USA, or China-derived isolates, indicating carry-over of the emerging ARGs from heavily industrialized countries and MSM-burdened (men who have sex with men) populations. The global burden of resistance GIs has increased sharply, especially after 2014. Emerging resistance mediators were most frequent (>80%) in human-associated *Escherichia coli* and *Klebsiella pneumoniae*. We infer ARGs horizontally propagate among Enteropathogens: informing treatment strategies and supporting policymakers in strengthening AMR-containment efforts utilizing the phylodynamics network.

**IMPORTANCE** The world is suffering from a high burden of MDR enteropathogens. Healthcare providers in low- and middle-income countries (LMICs) often face trouble finding effective drugs among the many antibiotics introduced in diarrheal treatment. Resistance-mediated drug inactivation is more rapid than the advent of new antimicrobials, leaving enteritis treatment on the edge. In Bangladesh, where one-third of users are self-prescribing antibiotics and thousands are dying due to resistance-related treatment failure, phylogenomic evidence of AMR transmission root is scarce. Therefore, investigating the resistomes of MDR-*Shigella*, the leading cause of diarrheal deaths in Bangladesh, is crucial. We identified several emerging resistance mediators and their phylogenetic links to global entities, which is significant for improving shigellosis treatment and enhancing AMR containment strategies. Understanding the MDR mechanism in *Shigella* will help physicians choose effective drugs and anticipate resistance-mediated changes

Address correspondence to Zhahirul Islam, zislam@icddrb.org.

The authors declare no conflict of interest.

See the funding table on p. 15.

in treatment approaches; the spatiotemporal phylodynamics of AMR mediators aid policymakers in setting effective checkpoints in the AMR transmission network.

**KEYWORDS** *Shigella*, multi-drug resistance, whole-genome sequencing, resistome phylodynamics, mobile genetic elements, conjugative plasmids, resistance genomic islands

*S*higella is a major causative agent of bacillary dysentery and diarrheal morbidity-mortality, particularly affecting low- and middle-income countries (LMICs) (1, 2). According to the Global Antimicrobial Surveillance System (GLASS), *Shigella* ranks as the second most deadly pathogen of bacterial diarrhea, causing 3%–6% of the diarrheal burden globally (1, 3). While shigellosis often resolves on its own, antimicrobial therapy is commonly used to manage *Shigella* infection (4). However, the emergence of antimicrobial resistance (AMR) has posed a serious challenge to effective shigellosis treatment (5). Mainstream treatment options are limited, and the prevalence of multi-drug resistance (MDR) further restricts available choices (6–8).

The rapid spread of resistance genes among bacterial populations is primarily attributed to mobile genetic elements (MGEs), such as plasmids, transposons, and integrons, which facilitate the efficient transfer of resistance genes between bacteria. Despite some studies focusing on specific drug resistance mechanisms, a comprehensive understanding of the entire resistomes of the MDR-*Shigella* genomes and their global epidemiology and dynamics remains largely unclear. The earliest documented case of multi-drug resistance in *Shigella* dates back to the 1950s in Japan when the bacterium became resistant to chloramphenicol, tetracycline, streptomycin, and sulfonamide drugs (9, 10). After developing resistance to aminoglycosides and sulfonamide-trimethoprim in *Shigella*, ciprofloxacin has long been recommended by WHO as the primary treatment option where ceftriaxone and azithromycin stood in the second line (3). Since then, *Shigella* has developed resistance to most of the antibiotics, including those traditionally used to treat shigellosis, such as fluoroquinolones, macrolides, and third-generation cephalosporins (3GCs) (6, 11). Resistance to ciprofloxacin was mostly induced by several mutations in *gyrA* and *parC* genes (12, 13). Macrolide resistance mostly takes place with the presence of the plasmid-borne *mph*A gene. The pathogenic macrolide-resistant pKSR100 type plasmid was reported to be responsible for intercontinental dissemination of macrolide resistance in men who have sex with men (MSM) patients (14, 15). Recently in Bangladesh, these potential pKSR100-type plasmids were reported to be driving macrolide resistance in *Shigella,* especially after 2014 (16, 17). Cephalosporin resistance is mostly anchored by beta-lactamase-producing genes like $bla_{TEM}$, $bla_{CTXM}$, or $bla_{OXA}$ genes (18, 19). In addition, the location of AMR factors in the genome is crucial to determine their transfer potential.

Global studies on AMR transmission routes in *Shigella* can be settled on a postulation that the men who have sex with men (MSM) population is the single most primordial source of emerging AMR-burden in *Shigella* globally (14, 15, 20–23). Outbreaks of emerging MDR-*Shigella* among MSM have been documented since the 1970s, particularly in Europe (22). The global burden of azithromycin and 3GC resistance in *Shigella* can be correlated with the prevalence of the MSM population, while Latin American and European countries have the highest MSM burden (3.77% and 2.11%, respectively) according to WHO (24).

Whole-genome sequencing (WGS) and phylogenomics have significantly enhanced pathogen surveillance and facilitated detailed monitoring of pathogens. The dynamics of MDR-*Shigella* resistomes remain poorly understood, especially in LMICs like Bangladesh. The lack of such studies in Bangladesh, where *Shigella*-associated diarrhea imposes a significant health burden, underscores the urgent need for a comprehensive understanding of resistome profiles, epidemiology, and dynamics.

Access to enriched databases for pathogens, MGEs, and ARGs along with efficient bioinformatic pipelines has provided a new framework for investigating MDR-resistome

evolution (25, 26). Therefore, gaining clear insight into these aspects is crucial for developing targeted interventions to control the spread of resistance among *Shigella*.

This study aims to evaluate resistome profiles and depict the worldwide phylodynamics of MDR-*Shigella* resistomes through WGS-based approaches. Consequently, the result of this study could enhance comprehension of the emergence and AMR dissemination network in MDR-*Shigella*; this may potentially aid the development of regional treatment guidelines for shigellosis.

## RESULTS

### Susceptibility phenotypes of multi-drug-resistant *Shigella* spp

Each of the 11 studied *Shigella* spp. was resistant to five or more antibiotic classes (Ab-classes), thus phenotypically MDR. They were resistant to most of the first- and second-generation antibiotics of seven Ab-classes except carbapenem, monobactum, and phenicols which were conferred moderate resistance in *Shigella*. MDR-*Shigella* had a large spectrum of resistance to the all-generation antibiotics including major treatment options (ciprofloxacin, azithromycin, and ceftriaxone/cefixime) as well as some aminoglycosides and sulfonamides-trimethoprim (Supplementary file S1). *Shigella* was moderately resistant to 3GCs, a potential treatment alternative for shigellosis. MDR-*Shigella boydii* Z12931 was resistant to imipenem, one of the most potential carbapenem antibiotics. *Shigella flexneri* Z13032 was resistant to chloramphenicol, an important phenicols. MDR *S. sonnei* (Z12947, Z13154, and Z13254) and *S. flexneri* Z13032 were resistant to azithromycin, ciprofloxacin, and all four 3GCs (Fig. 1), whereas the MDR *S. boydii* Z12959 were resistant to azithromycin, ciprofloxacin and 2 of the four 3GCs (cefotaxime and cefixime). The aminoglycoside drugs streptomycin and amikacin were resistant in all MDR-*Shigella*.

### Pangenome analysis and plasmid profiling

We found a large portion of the pangenome to be soft core and accessory genome which mainly consists of MGEs like plasmids and transposons. Species-wise phylogenetic clustering was obvious in the accessory genomes (Fig. 2). All MDR-*Shigella* possessed conjugative, mobilizable, and non-mobilizable plasmids; conjugative MDR plasmid pAA282 (82–107 kbp) or pAA338 (85–95 kbp) were identified in every MDR isolates except *S. flexneri* Z13145 and *S. boydii* Z12985 (Table 1). The conjugative pAA282 plasmids were MDR and appeared in IncI-γ/K1 (in 3 *S. sonnei* and *S. boydii* Z12959), IncK2/Z (*S. sonnei* Z12965) or IncI1/B/O (in *S. boydii* Z12931) plasmid types. Each *S. sonnei* possessed pAA282 MDR-conjugative plasmid. The other conjugative and MDR plasmid pAA338 were IncFIA type and found in *S. flexneri* Z12966, *S. flexneri* Z13032, and *S. boydii* Z12959; however, it was absent in all *S. sonnei*. A large (>100 kbp) IncFIA-type non-resistant and non-conjugative plasmid pAB272 was identified in all *S. flexneri* and *S. boydii* genomes; only *S. sonnei* possessing the plasmid as a conjugative and resistant one (pZ13254_AB272) (Table 1). Four atypical non-mobilizable AMR plasmids were identified; pAC314 (26–29 kbp in 3 *S. sonnei* and 15 kbp in 1 *S. flexneri*), pAC239 (in *S. flexneri* Z12966 and *S. boydii* Z12959; 4.5 kbp), pAC293 (in *S. boydii* Z12959; 10.5 kbp), and pAF098 (in *S. flexneri* Z13032; 11 kbp). Interestingly, all conjugative plasmids were conferring resistance factors to multiple drugs.

### Resistome profiling

#### *Conjugative MDR plasmid with pAA338 backbone*

The pAA338-backbone plasmids were IncF-type and similar to the pKSR100 plasmid. In resemblance to the pKSR100 plasmid, they possessed the macrolide resistance *mphA*-cluster [*mphA-mrx-mph(R)A*-IS*6100*] neighbored with an *ermB* gene downstream and an extended-spectrum beta-lactamase (ESBL)-gene *bla*$_{TEM-1}$ upstream (Fig. 3). In addition, the pZ13032_AA338 harbored crucial ESBL-producing gene *bla*$_{CTX-M-15}$ in the

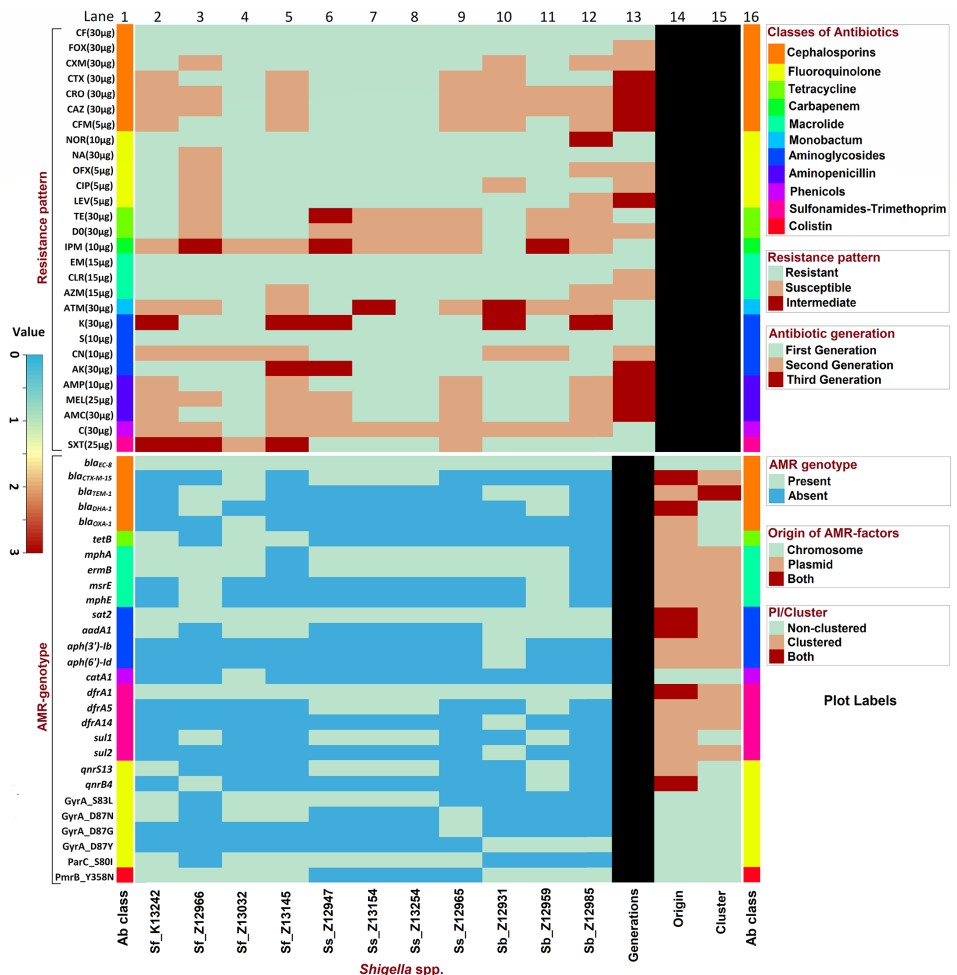

**FIG 1** AMR phenotype and genotype of 11 MDR-*Shigella*. Lanes 1 and 16: represent the classes of antibiotics; Lanes 2–12: MDR-*Shigella*; Lane 13: antibiotics' generations; Lane 14: plasmid or chromosomal origin of AMR-genes; Lane 15: status of genomic islands related to AMR genes. The black color represents no availability of data. Ab, antibiotic.

*wub*C-*bla*~CTX-M-15~-ISEcp1 gene cluster. Quinolone resistance genes *qnrS13* and *qnrB4* were present in pZ12959-AA338 and pZ12966_AA338 plasmids. The *S. flexneri*-borne plasmid pZ12966_AA338 also possessed beta-lactamase gene *bla*~DHA-1~ and sulfonamide-resistant *sul1* genes (Table 1; Fig. 3).

### Conjugative MDR plasmid with pAA282-backbone

Conjugative plasmids with pAA282 backbone possessed four different types of resistance gene clusters (Fig. 3). The *mphA*-resistance island in this plasmid was found in pZ12931_AA282, pZ12965_AA282, and pK13242_AA282; the cluster was similar to the *mphA*-cluster in pKSR100-type plasmids. The aminoglycosides, sulfonamides-trimethoprim, and Streptothricin N-acetyltransferase resistance genes originated from three different resistance gene clusters embedded in the pAA282-backbone plasmids. The first one, the *dfrA1-sat2-aadA1* resistance gene cluster was found in pZ12931_AA282, pZ12959_AA282, and pZ13242_AA282 plasmids; however, this cluster was missing in *S. sonnei*-originated plasmids. Another *dfrA14-sul2* cluster was found in all pAA282-backbone plasmids except pZ12931_AA282. The pZ12931_AA282 plasmid was carrying *bla*~TEM-1~-*aph(3′)-Ib-dfrA14-aph(3′)-Ib_sul2*-resistant genomic island (GI). An *aph-dfrA14-aph-sul2* genomic island was embedded over a Tn3 family transposon Tn2.

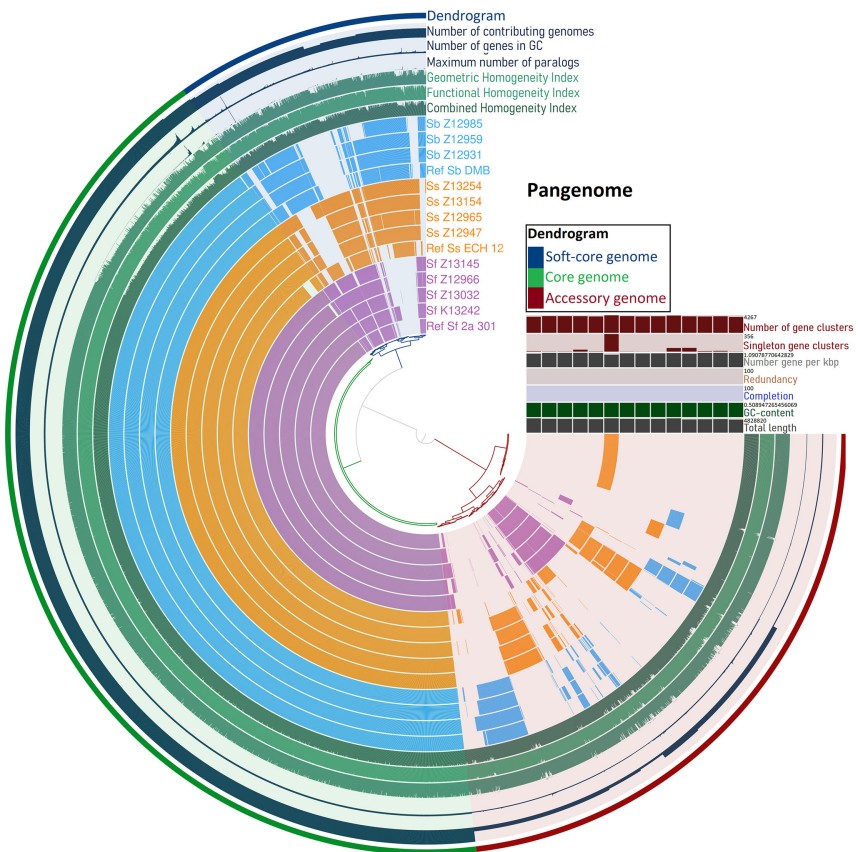

**Pangenome**

**Dendrogram**
- Soft-core genome
- Core genome
- Accessory genome

Dendrogram
Number of contributing genomes
Number of genes in GC
Maximum number of paralogs
Geometric Homogeneity Index
Functional Homogeneity Index
Combined Homogeneity Index

Number of gene clusters
Singleton gene clusters
Number gene per kbp
Redundancy
Completion
GC-content
Total length

**FIG 2** Pangenome profiling of 11 MDR-*Shigella* and 3 reference sequences: *S. flexneri* 2a strain 301, *S. sonnei* strain ECH 12S, and *S. boydii* strain DMB. A dendrogram was drawn to separate the core and accessory genomes. Color shades represent different species- purple = Sf = *Shigella flexneri*; Yellow = Ss = *Shigella sonnei*; Blue = Sb = *Shigella boydii*.

## ARGs associated with non-conjugable plasmids

The quinolone-resistant *qnrS13* gene was found in all pAC314-backbone plasmids. In *S. sonnei*, this plasmid possessed a macrolide-resistant *mphA*-gene cluster (pZ12947_AC314, pZ13154_AC314, and pZ13254_AC314) (Fig. 4A). Another atypical pAC239-backbone plasmid (pZ12966_AC239 and pZ12959_AC239) was harboring *mphE-msrE*-IS*482*-IS*6* ARG-cluster which is reported to be novel in *Shigella* (Table 1; Fig. 4B). It was also present in atypical non-mobilizable plasmids in three *S. sonnei*-Z12947 (pZ12947_AC314), Z13154 (pZ13154_AC314), and Z13254 (pZ13254_AC314) strains (Table 1; Fig. 4A). Atypical pAF098-backbone plasmid pZ13032_AF098 was conferring *bla*<sub>OXA-1</sub> and *tet*B resistance factors. (Table 1; Fig. 4C)

## Chromosomal ARGs

Quinolone-resistance-related mutations were the major chromosomal drug resistance mechanisms. We found multiple mutations in chromosomal genes of GyrA (S83L, D87N, D87G, and D87Y) and ParC_S80I proteins conferring resistance to fluoroquinolones. The GyrA_S83L and ParC_S80I mutations were found in all second-generation (ciprofloxacin and ofloxacin) and third-generation (levofloxacin) fluoroquinolone-resistant *S. flexneri* and *S. sonnei* but none of the *S. boydii*. In *S. boydii*, the GyrA_D87Y mutation was the key gene to confer resistance against second and third-generation fluoroquinolones. Chromosomal *bla*<sub>CTX-M-15</sub> gene-containing IS*1380-bla*<sub>CTX-M-15</sub>*-wbuC* ARG-cluster was identified in three of the MDR *S. sonnei* except *S. sonnei* Z12965. The *dfrA1-sat2-aadA1* resistance GI was found in the chromosome of *S. flexneri* Z13145, *S. flexneri* Z13032,

**TABLE 1** AMR-profiling of plasmids and chromosomes of MDR-*Shigella*[a]

| Strain | Chromosome/ plasmid name | Plasmid type | Plasmid mobility | Length | AMR genotype[b] |
|---|---|---|---|---|---|
| *S. flexneri* K13242 | Chromosome | NA | NA | 4,374,757 | **Chromosome:** *bla*$_{EC-8}$, GyrA_D87N, GyrA_S83L, ParC_S80I, GlpT_E448K, PmrB_Y358N, *tetB* |
| | pK13242_AA282 | IncK2/Z | Conjugative | 87,757 | **pK13242_AA282:** *dfrA1*, *sat2*, *aadA1*, *mphA*, *ermB* |
| | pK13242_AC314 | Atypical | Non-mobilizable | 15,248 | **pK13242_AC314:** *qnrS13* |
| | pK13242_AB272 | IncFIA | Mobilizable | 161,492 | |
| *S. flexneri* Z12966 | Chromosome | NA | NA | 4,344,520 | **Chromosome:** *bla*$_{EC-8}$, *dfrA1*, *sat2*, *emrE*, GlpT_E448K, PmrB_Y358N, *qacE*, *emrE* |
| | pZ12966_AA338 | IncFIA | Conjugative | 95,290 | **pZ12966_AA338:** *bla*$_{DHA-1}$, *bla*$_{TEM-1}$, *qnrB4*, *qacE*, *sul1*, *mphA*, *ermB* |
| | pZ12966_AC239 | Atypical | Non-mobilizable | 4,592 | **pZ12966_AC239:** *mphE*, *msrE* |
| | pZ12966_AB272 | Atypical | Non-mobilizable | 144,169 | |
| | pZ12966_AD168 | rc_2335 | Mobilizable | 6,978 | |
| | pZ12966_novel | IncFIA | Non-mobilizable | 5,341 | |
| *S. flexneri* Z13032 | Chromosome | NA | NA | 4392,855 | **Chromosome:** *bla*$_{EC-8}$, *dfrA1*, *sat2*, *aadA1*, GyrA_D87N, GyrA_S83L, ParC_S80I, *catA1*, GlpT_E448K, PmrB_Y358N |
| | pZ13032_AA338 | IncFIA | Conjugative | 91,501 | **pZ13032_AA338:** *bla*$_{CTX-M-15}$, *bla*$_{TEM-1}$, *mphA*, *ermB* |
| | pZ13032_AF098 | Atypical | Non-mobilizable | 10,994 | **pZ13032_AF098:** *bla*$_{OXA-1}$, *tetB* |
| | pZ13032_AB272 | Atypical | Non-mobilizable | 145,008 | |
| *S. flexneri* Z13145 | Chromosome | NA | | 4,377,831 | **Chromosome:** *bla*$_{EC-8}$, *dfrA1*, *sat2*, *aadA1*, ParC_S80I, GlpT_E448K, PmrB_Y358N, GyrA_D87N, GyrA_S83L, *tetB* |
| | pZ13145_AF098 | Atypical | Non-mobilizable | 4,604 | |
| | pZ13145_AB272 | IncFIA | Mobilizable | 162,968 | |
| *S. sonnei* Z12947 | Chromosome | NA | NA | 4,510,939 | **Chromosome:** *bla*$_{EC-8}$, *bla*$_{CTX-M-15}$, *ermB*, *dfrA1*, *sat2*, ParC_S80I, GlpT_E448K, GyrA_S83L, *qacEdelta1* |
| | pZ12947_AA282 | IncI-γ/K1 | Conjugative | 82,702 | **pZ12947_AA282:** *sul1*, *qacEdelta1*, *dfrA5* |
| | pZ12947_AC314 | Atypical | Non-mobilizable | 29,783 | **pZ12947_AC314:** *qnrS13*, *mphA* |
| | pZ12947_novel | IncFIC | Non-mobilizable | 18,769 | |
| | pZ12947_AA979 | Col156 | Mobilizable | 6,142 | |
| | pZ12947_AA974 | Col156 | Mobilizable | 5,241 | |
| *S. sonnei* Z12965 | Chromosome | NA | NA | 4,475,304 | **Chromosome:** *bla*$_{EC-8}$, *dfrA1*, *sat2*, ParC_S80I, GlpT_E448K, GyrA_D87G, GyrA_S83L, *acrF*, *emrD* |
| | pZ12965_AA282 | IncK2/Z | Conjugative | 97,425 | **pZ12965_AA282:** *mphA*, *ermB* |
| | pZ12965_novel | rc_2131 | Non-mobilizable | 9,515 | |
| | pZ12965_AA979 | Col156 | Mobilizable | 6,142 | |
| | pZ12965_AA974 | Col156 | Mobilizable | 5,241 | |
| *S. sonnei* Z13154 | Chromosome | NA | NA | 4,493,841 | **Chromosome:** *bla*$_{EC-8}$, *bla*$_{CTX-M-15}$, *ermB*, *dfrA1*, *sat2*, *qacEdelta1*, ParC_S80I, *acrF*, GlpT_E448K, *emrD*, GyrA_S83L |
| | pZ13154_AA282 | IncI-γ/K1 | Conjugative | 84,273 | **pZ13154_AA282:** *dfrA5*, *qacEdelta1*, *sul1* |
| | pZ13154_AC314 | Atypical | Non-mobilizable | 28,592 | **pZ13154_AC314:** *qnrS13*, *mphA* |
| | pZ13154_AA352 | rc_2131 | Non-mobilizable | 13,490 | |
| | pZ13154_AA979 | Col156 | Mobilizable | 6,142 | |
| | pZ13154_AA974 | Col156 | Mobilizable | 5,241 | |
| *S. sonnei* Z13254 | Chromosome | NA | NA | 4,544,640 | **Chromosome:** *bla*$_{EC-8}$, *bla*$_{CTX-M-15}$, *ermB*, *sat2*, *dfrA1*, ParC_S80I, *acrF*, GlpT_E448K, GyrA_S83L, *emrD* |
| | pZ13254_AA282 | IncI-γ/K1 | Conjugative | 85,039 | **pZ13254_AA282:** *sul1*, *qacEdelta1*, *dfrA5* |
| | pZ13254_AC314 | Atypical | Non-mobilizable | 26,923 | **pZ13254_AC314:** *qnrS13*, *mphA* |
| | pZ13254_AB272 | rc_2131 | Conjugative | 153,914 | |
| | pZ13254_AA979 | Col156 | Mobilizable | 6,142 | |
| | pZ13254_AA974 | Col156 | Mobilizable | 5,241 | |
| *S. boydii* Z12959 | Chromosome | NA | NA | 4,234,162 | **Chromosome:** *bla*$_{EC-8}$, *bla*$_{DHA-1}$, *qnrB4*, GlpT_E448K, PmrB_Y358N, GyrA_D87Y, *qacEdelta1*, *mphE*, *msrE* |
| | pZ12959_AA282 | IncI-γ/K1 | Conjugative | 85,488 | **pZ12959_AA282:** *dfrA1*, *sat2*, *aadA1* |
| | pZ12959_AA338 | IncFIA | Conjugative | 85,880 | **pZ12959_AA338:** *bla*$_{TEM-1}$, *qnrS13*, *mphA*, *ermB* |
| | pZ12959_AC293 | Atypical | Non-mobilizable | 10,618 | **pZ12959_AC293:** *sul1*, *qacEdelta1*, *dfrA5* |
| | pZ12959_AC239 | Atypical | Non-mobilizable | 4,592 | **pZ12959_AC239:** *mphE*, *msrE* |
| | pZ12959_AD446 | Atypical | Non-mobilizable | 13,777 | |
| | pZ12959_AB272 | Atypical | Non-mobilizable | 100,372 | |
| | pZ12959_AG294 | Atypical | Non-mobilizable | 21,974 | |
| *S. boydii* Z12931 | Chromosome | NA | NA | 4,226,967 | **Chromosome:** *bla*$_{EC-8}$, GlpT_E448K, PmrB_Y358N, GyrA_D87Y, *aph(3")Ib* |
| | pZ12931_AA282 | IncI1/B/O | Conjugative | 108,545 | **pZ12931_AA282:** *bla*$_{TEM-1}$, *dfrA1*, *sat2*, *aadA1*, *dfrA14*, *sul2*, *aph (6)Id*, *aph(3")Ib*, *mphA*, *ermB* |

**TABLE 1** AMR-profiling of plasmids and chromosomes of MDR-*Shigella*[a] (*Continued*)

| Strain | Chromosome/ plasmid name | Plasmid type | Plasmid mobility | Length | AMR genotype[b] |
|---|---|---|---|---|---|
| | pZ12931_AB272 | IncFIA | Mobilizable | 130,488 | |
| | pZ12931_AD446 | Atypical | Non-mobilizable | 13,780 | |
| | pZ12931_AG294 | Atypical | Non-mobilizable | 21,110 | |
| *S. boydii* Z12985 | Chromosome | NA | NA | 4,212,436 | **Chromosome:** *bla*$_{EC-8}$, *dfrA1*, *sat2*, *aadA1*, GlpT_E448K, PmrB_Y358N, GyrA_D87Y |
| | pZ12985_ AB272 | IncFIA | Mobilizable | 127,370 | |
| | pZ12985_ AD446 | Atypical | Non-mobilizable | 18,720 | |
| | pZ12985_ AG294 | Atypical | Non-mobilizable | 21,974 | |

[a]NA, not applicable.
[b]The boldface indicates the plasmid IDs and the other gene names belong to the respective plasmids.

and *S. boydii* Z12985; however, in all *S. sonnei* chromosomes, this cluster was present as *dfrA1-sat2* except *aadA1* gene (Table 1).

## Resistome dynamics

### Evolution and global dissemination pattern of pAA282 and pAA338 backbone MDR plasmids

NCBI-blast (blastn) for the IncK2/Z-type R-plasmid pK13242_AA282 resulted in a phylogenetic tree with several distinct phylogroups (Fig. 5A). The pAA282-backbone plasmid (2018) belonged to the PG2b_b2 phylogroup formed of USA- (2016–2018), South Korea- (2016-2017), and China- (2015) originated plasmids. The pAA282-plasmid first appeared in Bangladesh through an *E. coli* isolated in 1998 that was closely related to the USA- and China-originated plasmids (PG2b_a1 of Fig. 5A). This epidemic plasmid in *Shigella* was found in formerly reported from Hungary (1954) that showed phylogenetic relativeness to the plasmids isolated after 2000 from the USA (*E. coli*), Spain (*Shigella* and *Citrobacter*), Poland (*Klebsiella*), South Korea (*E. coli*), Germany (*E. coli*), and Australia (*E. coli*). However, most of the pAA282 plasmids were identified from China, South Korea, and the UK (Fig. 5A). On the other hand, the emerging pKSR100-type R-plasmid with pAA338 backbone, plasmid pZ13032_AA338 resulted in 280 hits and a tree with discrete phylogenetic clusters (Fig. 6A). The query *Shigella* plasmid pZ13032_AA338 formed PG2_a1 phylogroup (87% coverage with >99.9% identity) with an *E. coli* plasmid pEc2-50748 (CP104119.1) isolated from the USA (2013) and a *S. flexneri*

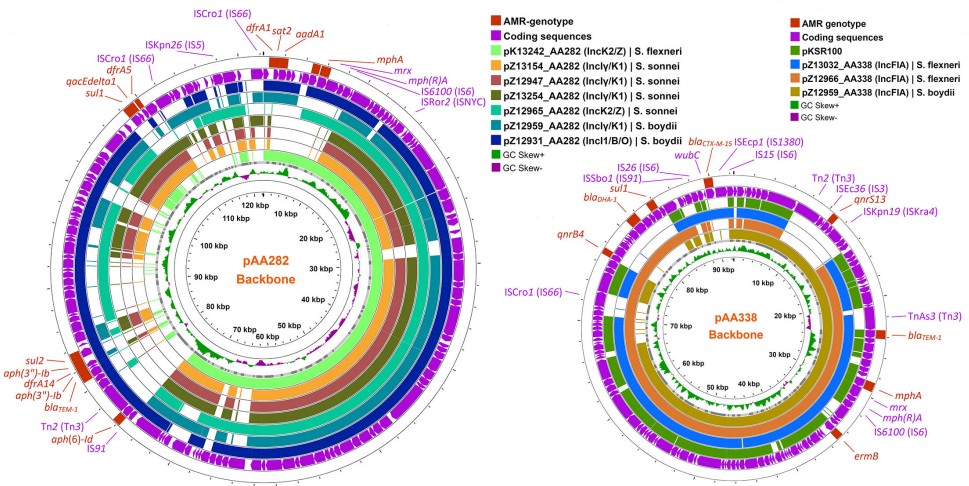

**FIG 3** AMR gene mapping of the two MDR conjugative plasmids, pAA338 and pAA282. AMR genes were labeled red, and the resistance-associated/neighboring non-AMR genes were labeled blue. HGT, horizontal gene transfer.

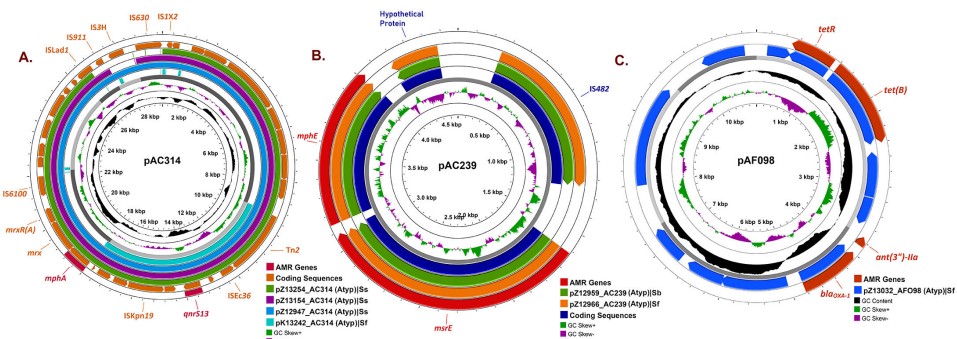

**FIG 4** AMR gene location in non-conjugable plasmids. (A) pAC314 plasmid-carrying *mphA*-GI as well as *qnrS13* gene. (B) pAC293 plasmid in carrying new macrolide resistance genes. (C) Beta-lactamase encoding *bla*$_{OXA-1}$ gene in pAF098 plasmid.

plasmid isolated from the UK (2021). The earliest member of phylogroup PG2_a1 was *S. flexneri* plasmid 981p2 (CP012139.1) isolated from China in 1998. In addition to this, *E. coli* plasmid pMTY2330_IncFII (AP026473.1), *E. coli* plasmid pHK23a (JQ432559.1), *E. coli* plasmid pSUH-2 (CP041339.1), and *K. pneumoniae* plasmid 31 (LT968717.1) from Japan, Hong-Kong, Singapore, and China in 2005, 2008, 2008, and 2009, respectively, were the other ancestral plasmids of this phylogroup (Fig. 6A). However, the pAA338 plasmids in Bangladesh were abundant in *Klebsiella pneumoniae* and *E. coli* (phylogroup PG2_b2) isolated from wound swab and blood samples which were phylogenetically linked (>85% coverage with >99.9% identity) with Myanmar-, India-, Nepal-, Pakistan-, and South Korea-originated plasmids. The phylogenetic also indicated the oldest signatures of the pAA338 phylogeny which were found in Brazilian *E. coli* strain BH100 and BH100L (phylogroup PG1), isolated from urine samples in 1974 (Fig. 6A). Overall, *Shigella*-associated pAA33-backbone plasmids were predominant in the UK, Spain, Australia, Switzerland, and China. *E. coli* was the major reservoir species for both pAA282 (84%) and pAA338 (47%), whereas *Shigella* and *Klebsiella* were the second leading reservoirs (pAA282: 12% and 20%; pAA338: 20% and 26%, respectively) (Fig. 5B and 6C). Humans were the predominant host for both the pAA282 and pAA338 plasmids (72% and 81%) followed by animals (15% and 10%) and water sources (9% and 5%), respectively (Fig. 5C and 6B). The rate of isolation of these conjugative plasmids was found to be dramatically increased after 2014 (Fig. 5D and 6D)

### Global phylodynamics of *mphA* and *bla*$_{CTX-M-15}$-associated ARG-cassettes

Blastn performed for the ARG cassettes in PLSDB, a dedicated plasmid database, resulted in almost similar results to the MDR-R-plasmids with few exceptions in the species abundance (Fig. 7A through D and 8A through D). The IS*26-mphA-mrx-mph(R)A*-IS*6100* cluster was found in 516 plasmids worldwide, *E. coli* was the predominant organism (48%) followed by *Klebsiella* (31%) with humans as the major host (74%) and animal in the second (9%) (Fig. 7C and D). The earliest wave of macrolide-resistant gene cluster was found to be in the USA, Mexico, and Brazil from 2000 to 2009. The second wave of the earliest period was from Europe (2005–2012) and South-East Asia (2008–2018) (Fig. 7A). A total of 514 hits of IS*1380-bla*$_{CTX-M-15}$-*wbuC* GI were found from PLSDB. Unlikely the plasmid distribution, this 3GC-resistant *bla*$_{CTX-M-15}$ GI was mostly found in *Klebsiella* (48%) rather than *E. coli* (38%); however, the human host was found still predominant (77%) (Fig. 8B and C). The first wave of the 3GC-resistant IS*1380-bla*$_{CTX-M-15}$-*wbuC* ARG cluster was found to be from Canada, the USA, and Brazil (1999–2009). Like the *mphA* GI, Europe stands second in the geographical timeframe (2001–2008) followed by South-East Asia (China, Hong Kong, Indonesia, South Korea, and India) (2006–2010) and Australia (2008) (Fig. 8A). Overall, South-East Asia persists as the major hotspot in

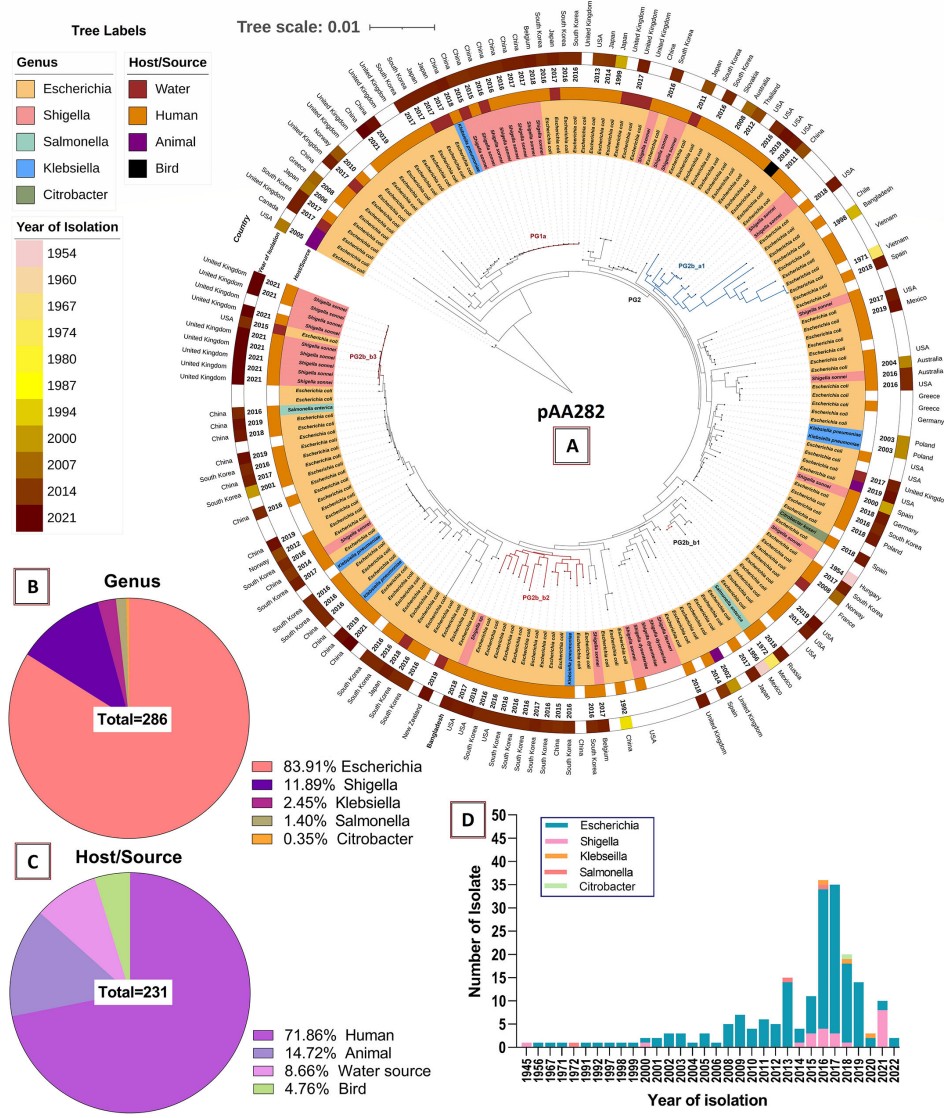

**FIG 5** Global phylodynamics of the IncI-type conjugative-plasmid pAA282 in MDR-*Shigella* in Bangladesh. (A) Global phylogeny of plasmid pK13242_AA282 in terms of species, year of isolation, isolation source, and associated bacterial genera. (B) Percentage distribution of genera possessing pAA282. (C) Percentage distribution of isolation sources possessing pAA282. (D) Temporal abundance boost of pAA282.

all the cases, particularly China. The incidence rate increased sharply after 2014, like the previous results from plasmid epidemiology (Fig. 7B and 8B).

## DISCUSSION

In this study, we assessed the resistomes of clinical MDR-*Shigella* and evaluated the phylodynamics of the AMR-related genomic entities like plasmids, and GIs. We report several emerging AMR-GIs causing the AMR-boost in Bangladesh including a third-generation cephalosporin-resistant *bla*$_{CTX-M-15}$ gene cassette and macrolide-resistant *mphA* gene cassette in a single pKSR100-type plasmid. It was indicated that the major emerging ARGs in MDR-*Shigella* were MGE borne and evolved through global AMR exchange. Furthermore, our analysis suggests that South-East Asia serves as a hotspot for antimicrobial resistance, with notable contributions of the developed countries in the American and European regions to the evolution and dissemination of ARGs.

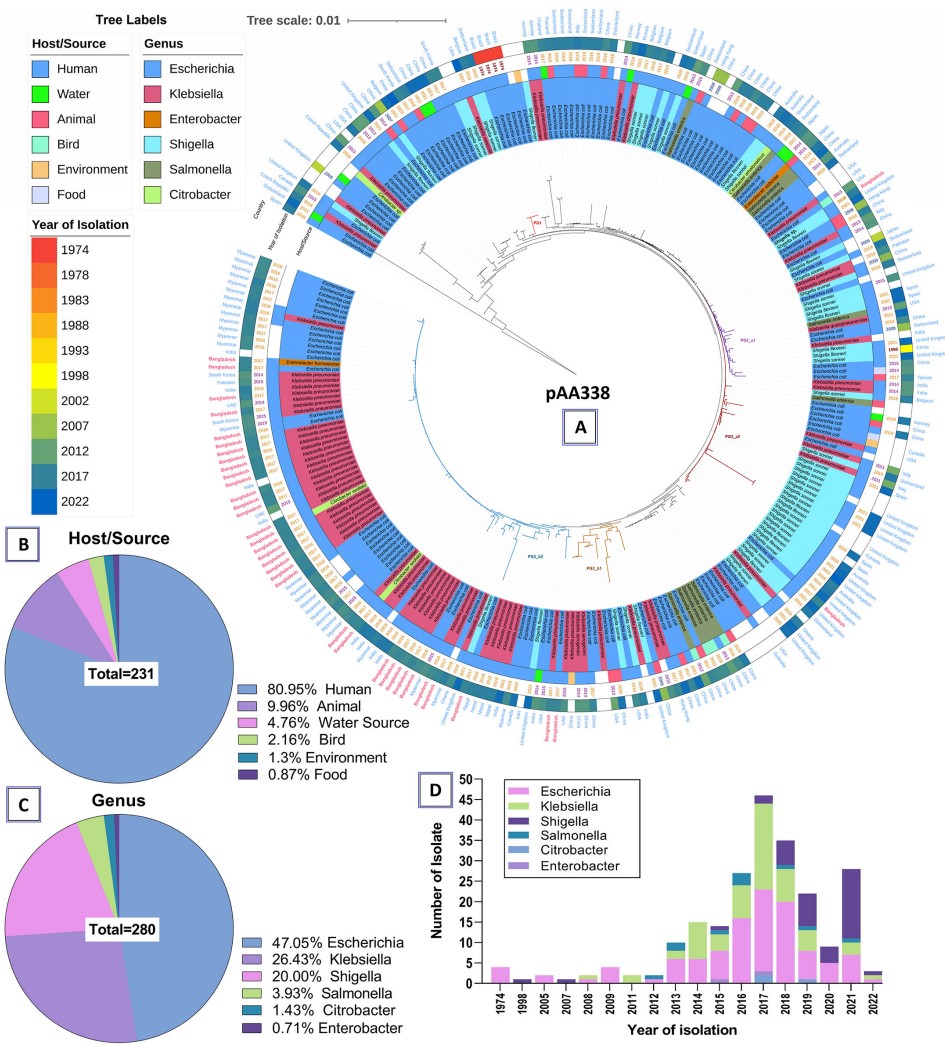

**FIG 6** Global phylodynamics of the IncF-type conjugative-plasmid pAA338 in MDR-*Shigella* in Bangladesh. (A) Phylodynamics of pAA338 presented with information like, year of isolation, source of isolation, and bacterial genera. (B) Percentage distribution of genera possessing pAA338. (C) Percentage distribution of isolation sources possessing pAA338. (D) Temporal abundance boost of pAA338.

Bangladesh ranks among the nations with the most significant burden of AMR globally. A recent study conducted by icddr,b, covering 883 urban and 1,263 rural cases of shigellosis spanning the past two decades (2001–2020), highlights the emergence of multidrug-resistant *Shigella* strains in Bangladesh (27). They showed a consistent percentage (~27%) of MDR-*Shigella* from 2006 to 2015, which had increased by over 55% in 2020. A sharp increase in ceftriaxone resistance in *Shigella* was found after 2014 and onwards in another study (17, 27). Therefore, it can be convincingly inferred that new AMR-triggering factors have been reinforced during 2015 and onwards. Therefore, we targeted MDR-*Shigella* isolated from the crucial turnover period (2015–2016) and undertook a robust genomics-based approach to uncover the MDR-mediating genetic makeovers with their root of transmission.

A pKSR100-like plasmid with pAA338-backbone was the most crucial entity facilitating drug resistance in *Shigella*. This MDR plasmid was identified in three of the five MDR-*Shigella*. In addition, ESBL-producing *bla*$_{CTX-M-15}$ gene containing *bla*$_{CTX-M-15}$-IS*1380*-*bla*$_{CTX-M-15}$-*wbuC* GI was found to be the macrolide resistance factors (*mphA*-gene cluster and the *ermB* gene), which is a very first report from Bangladesh. The pKSR100 plasmid

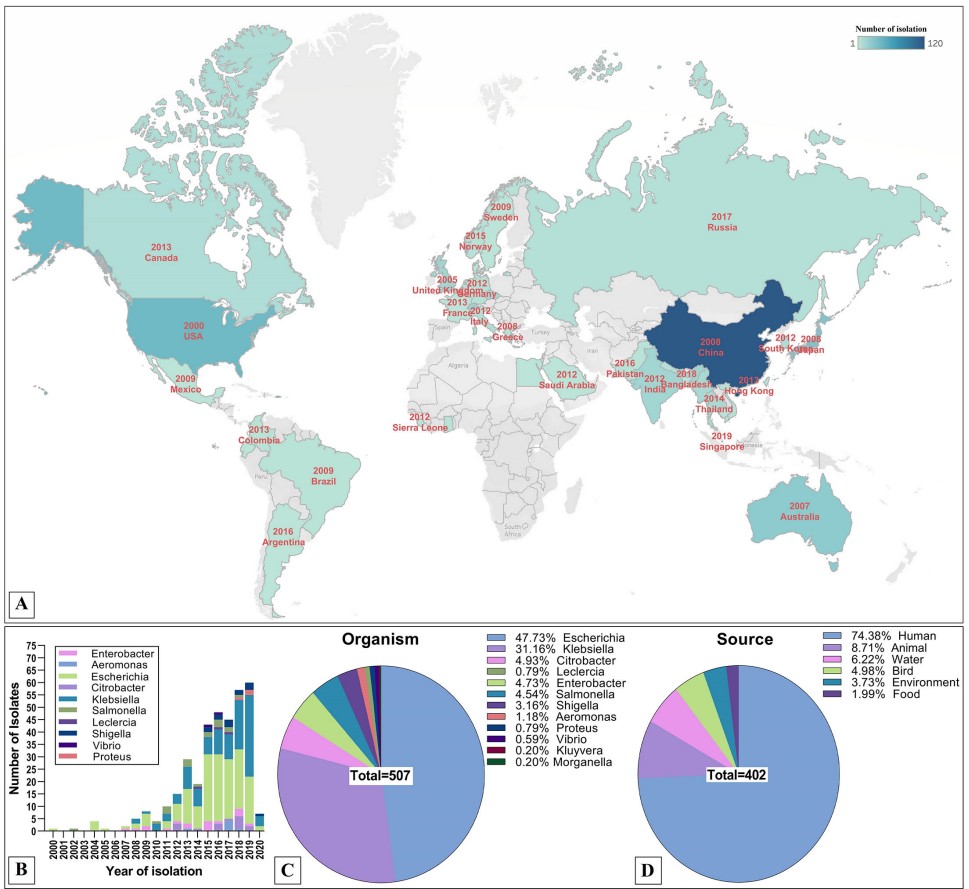

**FIG 7** Global distribution of macrolide-resistant IS*26-mphA-mrx-mph(R)A*-IS*6100* GI. (A) Country-wise number of isolations and the first year of isolation. (B) number of species isolations per year. Percentage distribution of source organism (C) and source host (D) possessing IS*26-mphA-mrx-mph(R)A*-IS*6100*.

has been known for its robust role in macrolide resistance (possessed *mphA*-gene cluster and the *ermB* gene): demonstrated high genetic adaptability and capacity to disseminate emerging AMR-factors among species and continents (5, 14, 23, 28). Incorporation of pKSR100-like plasmids with *bla*$_{CTX-M}$ genes has previously been reported to be associated with MSM populations in Europe, North America, and Australasia (14, 15, 20–23). The predominance of the pKSR100 plasmid-borne *bla*$_{CTX-M-15}$ gene in *S. sonnei* was shown in England and Wales (29). Our study also indicated the association of these plasmids with the USA and European-based isolations (Fig. 6 and 8); therefore, we designate this plasmid as the key genomic entity regarding the rising antibiotic resistance.

The MDR conjugable plasmid with pAA282 backbone was prevalent which carried further strength to AMR-dissemination potential in *Shigella*. This plasmid possessed several drug-resistant GIs against all potential drug classes. A similar plasmid has been reported before 2010 in European and Australian links which was carrying *aph* (6)Id-*aph* (3)Ib-*sul2-tet*A gene cluster along with ESBL genes and *mphA*-gene cluster altogether (30). The pAA282 plasmids in this study also carry the ESBL-producing *bla*$_{TEM-1}$ gene. The phylodynamics of this plasmid were slightly different from the pAA338-backbone plasmid; it also showed significant phylogenetic relativeness to China and South Korean isolates along with the American-European distribution. Due to the frequent isolation rate (7 in 11 MDR-*Shigella*) and high ARG burden, we indicate this plasmid as one of the major and emerging AMR mediators.

The AMR transmission potential may further be intensified due to the high abundance of drug-resistant non-conjugable plasmids and AMR-gene-associated

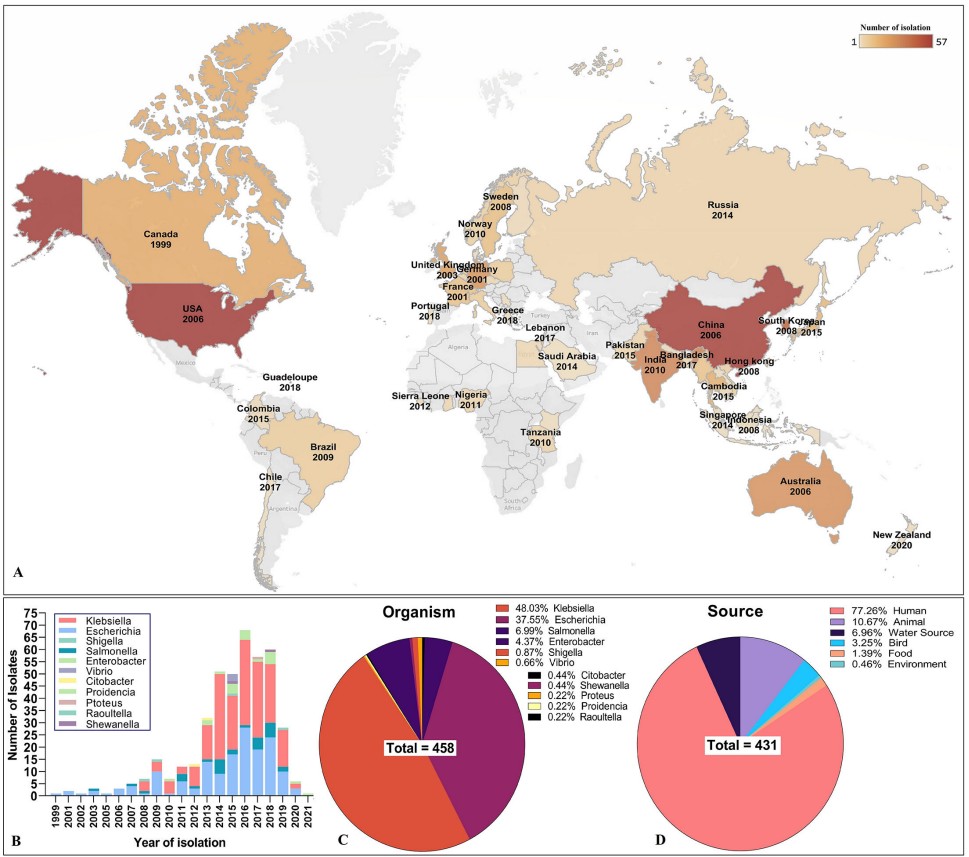

**FIG 8** Global distribution of macrolide-resistant IS*1380*-*bla*$_{CTX-M-15}$-*wbuC* ARG-cassette. (A) Country-wise number of isolations and the first year of isolation. (B) number of isolations with time. Percentage distribution of source organism (C) and source host (D) possessing IS*1380*-*bla*$_{CTX-M-15}$-*wbuC*.

transposons, integrons, and ISs. The non-conjugable plasmids can act mobile while using the transmission machinery from neighboring conjugable plasmids. The studied MDR-*Shigella* possessed multiple non-conjugable resistant plasmids neighbored by conjugable plasmids; therefore, the potential association of non-conjugable plasmids can be strongly inferred. We found In*2*/Tn*7*, IS*26*, IS*6100*, IS*91*, IS*482*, and IS*1380* were associated with the ARGs. Previous studies reported the association of In*2*/Tn*7*, IS*1*, IS*2*, IS*4*, IS*26*, IS*15*, IS*600,* and IS*911* with AMR in *Shigella* (22, 31). Among the ISs, IS*26* possesses the greatest effect in terms of AMR boost. Although IS*26* exerts no proven threat by itself, it can function as the anchor for further IS*26* insertion; thus, indirectly inviting the ARGs to be incorporated (16, 32). Overall, ISs favor the fluidity of pathogen genomes and may add a layer of strength to horizontal-transmission potential (33)

We highlight the MSM population bias for these emerging MDR-MGEs presumably regulates the leaderboard of the global scenario of shigellosis (15, 21, 22). Our findings suggest that MDR conjugable plasmids might be spawned in the MSM-burdened population in America, Europe, and China, and later transmitted internationally. However, due to poor health management systems and resource-limited socio-economic settings in South and East-Asian countries, these AMR-encoding MGEs spread rapidly. Our postulations were concordant with several global studies (22, 34, 35).

This study implies significant potential in terms of AMR-containment strategies, particularly in Bangladesh. It indicates a possible crisis of shigellosis treatment options near future and may aid physicians in prescribing drugs prudently. It also demands a particular focus on searching for alternatives. In addition, this study provides an excellent insight into global resistome-phylodynamics which can maximize global efforts to control AMR. However, insignificant data from Central Asia and Sub-Saharan Africa and a

lack of surveillance data from Bangladesh remain the limitations of this study. Therefore, all-country-in genomics surveillance of MDR-*Shigella* remains essential to complete the big picture of the global AMR network.

## MATERIALS AND METHODS

### Identification and isolation of *Shigella* strains

Eleven MDR-*Shigella* spp. investigated in the study were isolated from diarrheal patients admitted to the Dhaka Hospital of the International Centre for Diarrhoeal Disease Research, Bangladesh (icddr,b) between 2017 and August 2018. Standard microbiological culture methods like the aerobic culture on MacConkey agar (Difco) and subsequent 18-h incubation at 37°C were used to isolate the MDR-*Shigella* strains (36). The 11 *Shigella* strains (4 *S. flexneri*, 4 *S. sonnei,* and 3 *S. boydii*) were serologically confirmed by slide agglutination tests applying commercially available antiserum kits (Denka Seiken, Tokyo, Japan) (37).

### Antimicrobial susceptibility testing

The eleven *Shigella* spp. were subjected to antimicrobial susceptibility test (AST) against 28 antibiotics from 10 different drug groups, for example, fluoroquinolone ($n = 5$), macrolide ($n = 3$), cephalosporin ($n = 7$), tetracycline ($n = 2$), aminoglycosides ($n = 4$), carbapenem ($n = 1$), monobactams ($n = 1$), sulfonamides-trimethoprim ($n = 1$), phenicols ($n = 1$), and aminopenicillin ($n = 3$) (Supplementary file S1). AST of *Shigella* spp. was performed through a disk diffusion test using a commercially available antibiotic disk (Oxoid, Basingstoke, United Kingdom) following the methodology described elsewhere (36). *Escherichia coli* (ATCC 25922) was used as the negative control for the disk diffusion tests. The susceptibility was determined according to the susceptibility testing guidelines of the Clinical and Laboratory Standards Institute (CLSI Supplement M100S, 26th ed.) (38).

### Genomic DNA extraction, sequencing, and annotations

MDR-*Shigella* were inoculated in Luria-Bertani broth and incubated overnight at 37°C. Quality genomic DNA was ensured for WGS using the procedure described elsewhere (39). Next-generation sequencing (NGS) of the *Shigella* genomes was carried out on the Illumina MiSeq platform at the icddr,b Genomics Centre to generate 300 bp paired-end reads. The sequencing, quality control, assembly, and annotation methods were described previously (36). Assembled genomes were publicly deposited in the NCBI GenBank database under the BioProject accession numbers PRJNA693631, PRJNA694802, PRJNA698078, and PRJNA698772. The genome sequences were then annotated using NCBI Prokaryotic Genome Annotation Pipeline v5.0 (40–42). Additional annotations were performed using Prokka v1.14.5 and RAST web tool (43, 44).

### Pangenome analysis, plasmid sequence identification, and characterization

Pangenome was constructed from the 11 MDR-*Shigella* genomes and three reference genomes (*Shigella flexneri* 2a str. 301, *Shigella sonnei* strain ECH + 12, and *Shigella boydii* strain DMB SH136) using Anvi'o v7.1 (45). Individual plasmid sequences and chromosomal sequences were separately identified and typed in MOB-suite v3.1.0 (46). We also assembled plasmids from raw fastq sequences using Plasmid SPAdes v3.14.1 (47). The mobilization potential of the plasmids was also predicted in the MOB-suite v3.1.0 (46, 48). All the resulting plasmids and chromosomal sequences were then annotated using Prokka v1.14.5 and the RAST web-tool (43, 44). The plasmids were mapped and visualized using Proksee (49).

## Resistome profiling

We applied NCBI-AMRfinderPlus v3.10.5 (50) primarily and the outputs were further validated by comparing the AMR-gene annotations from Abricate (https://github.com/tseemann/abricate) and CARD (Perfect and strict hits) which also resulted in similar outputs (51). Heat plots were prepared using the Heatmaply package from R-repository (52).

## Dynamics and global phylogeny analysis

The two conjugative plasmids (pAA282 and pAA338) were blasted against the NCBI GenBank database. We could confirm bacterial species ($n = 286$), year of isolation (YI) ($n = 202$), source/host ($n = 231$), and country of isolation ($n = 239$) after mining the accessions provided with the pAA282-plasmid blast results. In the case of pAA338-plasmid blast, the source bacterial species ($n = 280$), year of isolation ($n = 227$), source/host ($n = 231$), and country of isolation ($n = 241$) were obtained by NCBI-data mining. The PLSDB v2021_06_23_v2 was used to find the distribution of AMR-gene clusters through phylogenetic analyses. We used the BLASTn search option with ≥80% sequence coverage and ≥95% identity. The resultant data sets were curated with the respective accession numbers. In addition, country, date, and source of isolation were incorporated from NCBI GenBank in the final data set. The phylogenetic trees were visualized and annotated in the iTOL tree preparation tool (53). Tableau's public version was used to prepare the global maps.

## ACKNOWLEDGMENTS

This research activity was primarily supported by activity no. ACT-00300 through icddr,b. It was partially supported by the Fogarty International Center, National Institute of Neurological Disorders and Stroke of the National Institutes of Health, USA, under Award Number K43TW011447 which was bestowed to ZI. icddr,b acknowledges with gratitude the commitment of the Government of The People's Republic of Bangladesh to its research efforts and gratefully acknowledges Canada for its unrestricted support.

Z.I. and A.A. conceptualized the study. Research methodology and execution plan were theorized and standardized by A.A. Z.I. provided the funding and capacity support. Laboratory experiments and data acquisition were performed by A.A., R.B., and S.N.F. Bioinformatics analysis of sequence data was done by A.A. while A.J.N. and G.M. assisted data analyses. R.B., S.N.F., A.A., A.J.N., G.M., R.B., and S.N.F. contributed to data mining. A.A. interpreted the experimental and computational outputs and drafted the manuscript. The manuscript was exclusively scrutinized by Z.I., A.J.N., G.M., and S.H. and it was reviewed by all authors. The final manuscript was read and approved by all authors before submission.

## AUTHOR AFFILIATIONS

[1]Gut-Brain Axis Laboratory, Infectious Diseases Division (IDD), icddr, b, Dhaka, Bangladesh
[2]Department of Genetic Engineering and Biotechnology, East West University, Dhaka, Bangladesh

## AUTHOR ORCIDs

Asaduzzaman Asad  http://orcid.org/0000-0001-6894-0910
Md. Abu Jaher Nayeem  http://orcid.org/0000-0002-2620-1762
Md. Golam Mostafa  http://orcid.org/0009-0008-4587-2239
Israt Jahan  http://orcid.org/0000-0001-9594-0705
Shoma Hayat  http://orcid.org/0000-0003-1064-9009
Zhahirul Islam  http://orcid.org/0000-0003-0935-8079

## FUNDING

| Funder | Grant(s) | Author(s) |
| --- | --- | --- |
| HHS \| NIH \| Fogarty International Center (FIC) | K43TW011447 | Zhahirul Islam |

## AUTHOR CONTRIBUTIONS

Asaduzzaman Asad, Conceptualization, Data curation, Formal analysis, Investigation, Methodology, Project administration, Software, Supervision, Validation, Visualization, Writing – original draft, Writing – review and editing | Md. Abu Jaher Nayeem, Data curation, Formal analysis, Methodology, Software, Visualization, Writing – review and editing | Md. Golam Mostafa, Formal analysis, Methodology, Software, Validation, Visualization, Writing – review and editing | Ruma Begum, Data curation, Formal analysis, Methodology, Project administration, Writing – review and editing | Shah Nayeem Faruque, Data curation, Formal analysis, Methodology, Writing – review and editing | Suraia Nusrin, Funding acquisition, Investigation, Project administration, Resources, Supervision, Writing – review and editing | Israt Jahan, Investigation, Project administration, Supervision, Writing – review and editing | Shoma Hayat, Data curation, Investigation, Project administration, Supervision, Writing – review and editing | Zhahirul Islam, Conceptualization, Funding acquisition, Investigation, Project administration, Resources, Supervision, Validation, Writing – review and editing

## DATA AVAILABILITY

WGS data, assembled genomes and annotations are available under the accession numbers PRJNA693631, PRJNA694802, PRJNA698772, PRJNA704496, and PRJNA698078. Wet laboratory test results are available in Supplementary file S1. Any additional information or analysis results are available from the corresponding author upon reasonable request.

## ETHICAL APPROVAL

All experiments were performed per relevant guidelines and regulations, and all participants gave their written informed consent before enrollment. The study was reviewed and approved by the Institutional Review Board (IRB) and the Ethical Committee of icddr,b, Dhaka, Bangladesh.

## ADDITIONAL FILES

The following material is available online.

### Supplemental Material

**Supplemental material (Spectrum01635-24-s0001.xlsx).** Antimicrobial Susceptibility Test (AST) results.

### Open Peer Review

**PEER REVIEW HISTORY (review-history.pdf).** An accounting of the reviewer comments and feedback.

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
