## [Reviewer comments · Microbiology Spectrum]

Microbiology Spectrum

Resistome phylodynamics of multidrug-resistant *Shigella* isolated from diarrheal patients

Asaduzzaman Asad, Md. Abu Jaher Nayeem, Md. Golam Mostafa, Ruma Begum, Shah Faruque, Suraia Nusrin, Israt Jahan, Shoma Hayat, and Zahirul Islam

Corresponding Author(s): Zahirul Islam, International Centre for Diarrhoeal Disease Research Bangladesh

Review Timeline:

Submission Date:	July 4, 2024
Editorial Decision:	August 19, 2024
Revision Received:	September 19, 2024
Accepted:	October 29, 2024

Editor: Cheryl Andam

Reviewer(s): The reviewers have opted to remain anonymous.

Transaction Report:

DOI: <https://doi.org/10.1128/spectrum.01635-24>

Re: Spectrum01635-24 (Resistome phylogenetics of multidrug-resistant *Shigella* isolated from diarrheal patients)

Dear Dr. Zahirul Islam:

Thank you for the privilege of reviewing your work. Below you will find my comments, instructions from the Spectrum editorial office, and the reviewer comments.

Revision Guidelines

Sincerely,
Cheryl Andam
Editor
Microbiology Spectrum

Reviewer #1 (Comments for the Author):

The manuscript of Asad and colleagues is a clear studies focused on the genomics characterization and resistome analysis of MDR *Shigella* strains in Bangladesh. The manuscript is well written, and the figures are clear. The authors have thoroughly analysed 11 strains, and the conclusions are justified by the data collected

Minor remarks:

L129: "Imipenem...was resistant in one". I find the sentence confusing, as bacteria show resistance to imipenem. Please clarify
L415: to investigate the resistome, different tools have been used. Did the results encompass all the predicted genes from the different databases, or only the genes found in common? Please clarify
Figure 3 and 4: Mapping of plasmids is not described in the Method section. Please specify the tools that were used to compare the sequences and generate the figures

Reviewer #2 (Comments for the Author):

The manuscript "Resistome phylodynamics of multidrug-resistant *Shigella* isolated from diarrheal patients" is well-written and includes interesting data. However, it needs further clarity and accuracy and hence a revision. Here, are my general and specific comments:

- The manuscript includes an interesting data set however, my major concern is that this interesting data has not been described well in the result section. While it is important to put your data into the global context, I think the text is distracted by NCBI data and has failed to describe the new genomes reported here thoroughly.
- The discussion is very long. It must be revised and turned into a much more concise section.
- Table 1: Please include chromosome lengths
- beta-lactam resistance genes. All beta-lactam resistance genes should be written with the "bla" section italic and the gene name "subscripted" e.g. in blaCTX-M-15, the "CTX-M-15" must be subscripted. Please fix throughout the manuscript including text, all Tables and Figures.
- What do you mean by "AMR transmission phylodynamics", could you please clarify? Is it intended to say "AMR transmission and phylodynamics"? Similarly the term "MDR-screened", I'd use a more informative term e.g. "screened for antimicrobial resistance genes/ARGs etc".
- Please be careful when using the term "gene cassette" as only a small fraction of antibiotic resistance genes are in fact in "Gene Cassettes". Gene cassettes refer to mobile genetic elements with specific characteristics such as the presence of attC (59-base element) and are often found in class 1 interns (regardless of whether they are in plasmids or chromosomes). That is to say, not all antibiotic resistance genes can be referred to as gene cassettes.
- Line 94: Please change "PKSR100" to "pKSR100" and fix it throughout.
- Line 122: You have to first establish they are MDR (or XDR and PDR) and then refer to them as such. Please consider revising this to follow a logical flow.
- Lines 128-129: The sentence "Resistance to 3GCs, one of the potential treatment alternatives for Shigellosis, was found moderately resistant in *Shigella*." Does not make any sense! Please revise!
- Line 129-130: Similarly, the sentence "Imipenem (IMP), one of the most potential Carbapenem antibiotics, was resistant in one MDR-*Shigella boydii* Z12931 strain." Has grammatical errors and does not make any sense! Please also revise!
- Line 132: This is confusing! Were they MDR or XDR? Each of these terms has specific definitions, please watch how they are used throughout the text! Please revise this sentence!
- Lines 131-137: Either use antibiotics' abbreviations or their full names (not both). Define their abbreviations in the method section so they don't have to be repeated throughout the text. Please revise!
- Line 140: Please stick to the more commonly (and understood) term "accessory genomes". Please avoid "cloud genomes".
- Line 146: What do you mean by "mostly" please be more specific (i.e. include a number, strain name etc.).
- Please watch the use of "mobilisable plasmids", which has a specific definition in plasmid biology. Mobilisable plasmids are often small plasmids that are co-transferred by large conjugative plasmids (or any other plasmids). They often encode a set of mobilisation genes e.g. relaxases such as trwB/C etc. Please note that with this definition a lot of non-conjugative plasmids might still be mobilisable. Please check!
- Lines 159-164: This section includes several vague statements. Either remove or improve this section by including more specific information.
- Please avoid using "underscore; _" to separate ARGs, it is very confusing.

- Line 187: "APH(3")-Ib" refers to the protein. Please use italic form of the gene version "*aph(3")-Ib*".
- Line 201: "most" is vague! Please be more specific!
- Line 203-204: You can't use gene names then refer to amino acid changes. Please change to Pr form (i.e. GyrA, ParC). Fix this throughout the text and Figures!
- Line 215: What do you mean by blast "resulted in 287 hits and 180 member distance tree"? Please clarify/revise!
- Line 262-263: Please revise "highly resistant IS26-mph(A)-mrx-mph(R)(A)-IS6100 ARG-cluster"!
- Line 336: When referring to the IS/Tn names, the numbers MUST be in italics.

Spectrum01635-24R1: Resistome phylodynamics of multidrug-resistant *Shigella* isolated from diarrheal patients

Response to Reviewers

Reviewer #1 (Comments for the Author):

Comment: The manuscript of Asad and colleagues is a clear study focused on the genomics characterization and resistome analysis of MDR Shigella strains in Bangladesh. The manuscript is well-written, and the figures are clear. The authors have thoroughly analyzed 11 strains, and the conclusions are justified by the data collected.

Response: We highly appreciate your positive comments on our submitted manuscript. We addressed all queries point-by-point in the revised manuscript.

Minor remarks:

Comment: L129: "Imipenem...was resistant in one". I find the sentence confusing, as bacteria show resistance to imipenem. Please clarify

Response: We revised the statement as "MDR-Shigella boydii Z12931 was resistant to Imipenem, one of the most potential Carbapenem antibiotics" as per the reviewer's suggestion (**Lines: 126-127**)

Query: L415: to investigate the resistome, different tools have been used. Did the results encompass all the predicted genes from the different databases or only the genes found in common? Please clarify

Response: The results presented in the manuscript encompass all the predicted genes and resistance-related mutations. The NCBI-AMRFinderPlus was the primary tool used for AMR gene identification. The other tools like Abricate (ncbi dataset) and CARD (perfect and strict hit) provided similar results except for the mutations. Resistance-associated mutations were assessed using the NCBI-AMRFinderPlus tool; however, we also revised the sentence (**Lines: 372-374**).

Comment: Figure 3 and 4: Mapping of plasmids is not described in the Method section. Please specify the tools that were used to compare the sequences and generate the figures

Response: Thank you for the comment. The plasmids were mapped and visualized using Proksee (<https://proksee.ca/>). We added the line in the methodology section in response to your comment (**Lines: 368-369**).

Reviewer #2 (Comments for the Author):

Comment: The manuscript "Resistome phylogenetics of multidrug-resistant Shigella isolated from diarrheal patients" is well-written and includes interesting data. However, it needs further clarity and accuracy and hence a revision. Here, are my general and specific comments:

Response: We highly appreciate your positive comment and keen review of this manuscript. We addressed your comments and suggestions and revised the current manuscript point-by-point.

Comment: The manuscript includes an interesting data set however; my major concern is that this interesting data has not been described well in the result section. While it is important to put your data into the global context, I think the text is distracted by NCBI data and has failed to describe the new genomes reported here thoroughly.

Response: Thanks for the comment and suggestion. In response to your comment, we revised the respective parts of the results section to keep it more concise and focused on the current MDR-*Shigella* genomes. The resistomes of the current MDR-*Shigella* have been thoroughly described under the "Resistome profiling" sub-heading (**Lines: 154-200**). In addition, we modified the "Resistome dynamics" section by focusing on the dynamics of new MDR-*Shigella*-related AMR mediators from our study. (**Lines: 203-2010, 214-220, 224-227, and 231-237**).

Comment: - The discussion is very long. It must be revised and turned into a much more concise section.

Response: Thank you for your suggestion. In response to your comment, we significantly reduced (~250 words) the length of the discussion in the current manuscript to be more concise and specific (**Lines: 258-325**).

Comment: - Table 1: Please include chromosome lengths

Response: We incorporated the chromosome lengths in **Table 1** in response to your comment (**Changes are highlighted**).

Comment: - beta-lactam resistance genes. All beta-lactam resistance genes should be written with the "bla" section italic and the gene name "subscripted" e.g. in bla_{CTX-M-15}, the "CTX-M-15" must be subscripted. Please fix throughout the manuscript including text, all Tables and Figures.

Response: Thank you for your keen observation. We corrected the naming of beta-lactam resistance genes throughout the manuscript in response to your comment.

Query: - What do you mean by "AMR transmission phylodynamics", could you please clarify? Is it intended to say "AMR transmission and phylodynamics"?

Response: Thanks for the query. It was a syntax error; it should be "AMR transmission and phylodynamics". We revised the current manuscript accordingly. (**Lines: 28-29**)

Comment: Similarly the term "MDR-screened", I'd use a more informative term e.g. "screened for antimicrobial resistance genes/ARGs etc".

Response: We appreciate your suggestion. We revised our manuscript accordingly and used "Shigella strains were screened for MDR phenotypes" replacing "MDR-screened". (**Line: 29**)

Comment: - Please be careful when using the term "gene cassette" as only a small fraction of antibiotic resistance genes are in fact in "Gene Cassettes". Gene cassettes refer to mobile genetic elements with specific characteristics such as the presence of attC (59-base element) and are often found in class 1 interns (regardless of whether they are in plasmids or chromosomes). That is to say, not all antibiotic resistance genes can be referred to as gene cassettes.

Response: Thank you for your concern. We checked and ensured the proper use of the term "gene cassette" throughout the revised manuscript. (**e.g. Lines: 32, 43, 183**)

Comment: - Line 94: Please change "PKSR100" to "pKSR100" and fix it throughout.

Response: Thank you for the keen review. We corrected the typo error and replaced PKSR100 with pKSR100 in the revised manuscript. **(Line: 91)**

Comment: - Line 122: You have to first establish they are MDR (or XDR and PDR) and then refer to them as such. Please consider revising this to follow a logical flow.

Response: We rewrote the sentence as "Each of the 11 studied *Shigella* spp. was resistant to five or more antibiotic classes (Ab-classes), thus phenotypically MDR" in the revised manuscript. **(Lines: 119-120)**

Comment: - Lines 128-129: The sentence "Resistance to 3GCs, one of the potential treatment alternatives for Shigellosis, was found moderately resistant in Shigella." Does not make any sense! Please revise!

Response: Thank you for your comment. We revised the sentence in the current manuscript "*Shigella* was moderately resistant to 3GCs, a potential treatment alternative for Shigellosis" **(Lines: 125-126)**.

Comment: - Line 129-130: Similarly, the sentence "Imipenem (IMP), one of the most potential Carbapenem antibiotics, was resistant in one MDR-Shigella boydii Z12931 strain." Has grammatical errors and does not make any sense! Please also revise!

Response: Thank you for the observation. The sentence has been modified in the revised manuscript **(Lines: 126-127)**

Comment: - Line 132: This is confusing! Were they MDR or XDR? Each of these terms has specific definitions, please watch how they are used throughout the text! Please revise this sentence!

Response: We used MDR to define *Shigella* which was resistant to more than 3 groups of antibiotics. Meanwhile, we used the term XDR to define strains susceptible to 2 or fewer drug groups. However, we uniformly used the term "MDR" in the revised manuscript to avoid confusion. **(Lines: 127-131)**

Comment: - Lines 131-137: Either use antibiotics' abbreviations or their full names (not both). Define their abbreviations in the method section so they don't have to be repeated throughout the text. Please revise!

Response: Thank you for your suggestion. We revised the current manuscript according to your instructions and used the full names of the antibiotics throughout the manuscript.

Comment: - Line 140: Please stick to the more commonly (and understood) term "accessory genomes". Please avoid "cloud genomes".

Response: Thank you for the suggestion. "Accessory genomes" have been uniformly used in the revised manuscript. **(Line 135 and Figure 2)**

Comment: - Line 146: What do you mean by "mostly" please be more specific (i.e. include a number, strain name etc.).

Response: We appreciate your constructive comments and efforts. In response, we rewrote the sentence in the revised version of the manuscript **(Line 141)**

Comment: - Please watch the use of "mobilisable plasmids", which has a specific definition in plasmid biology. Mobilisable plasmids are often small plasmids that are co-transferred by large conjugative plasmids (or any other plasmids). They often encode a set of mobilisation genes e.g. relaxases such as trwB/C etc. Please note that with this definition a lot of non-conjugative plasmids might still be mobilisable. Please check!

Response: I appreciate your concern and brief on this issue. We used the MOB-suite bioinformatics tool to define the mobility types of the plasmids. However, we further scrutinized the plasmids and their annotations to affirm the results as per your suggestion and found no aberration.

Comment: - Lines 159-164: This section includes several vague statements. Either remove or improve this section by including more specific information.

Response: Thank you for your suggestion. In response to your comment, we removed the section in the revised manuscript.

Comment: - Please avoid using "underscore;_" to separate ARGs, it is very confusing.

Response: Thank you for your suggestion. We avoided "underscore;_" to separate ARGs; instead used "hyphen; -" in the revised manuscript.

Comment: - Line 187: "APH(3)-Ib" refers to the protein. Please use italic form of the gene version "aph(3)-Ib".

Response: We appreciate your keen review. The naming has been corrected in the revised manuscript (**Line 175**).

Comment: - Line 201: "most" is vague! Please be more specific!

Response: We appreciate your constructive comments and efforts. In response, we rewrote the sentence in the revised version of the manuscript (**Line 190**).

Comment: - Line 203-204: You can't use gene names then refer to amino acid changes. Please change to Pr form (i.e. GyrA, ParC). Fix this throughout the text and Figures!

Response: We corrected the gene naming in the revised manuscript (**Lines: 191-195**). In addition, we made corrections according to your recommendation elsewhere in the manuscript.

Comment: - Line 215: What do you mean by blast "resulted in 287 hits and 180-member distance tree"? Please clarify/revise!

Response: We performed NCBI-blast (blastn) for the pK13242_AA282 plasmid which resulted in 287 matches worldwide. We constructed a phylogeny on the blast results, generating a phylogenetic tree with 180 distinguished leaves. However, in response to your suggestion, we revised the sentence for more clarity (**Lines: 205-206**).

Comment: - Line 262-263: Please revise "highly resistant IS26-mph(A)-mrx-mph(R)(A)-IS6100 ARG-cluster"!

Response: We revised the sentence as per your suggestion (**Lines: 242-243**).

Comment: - Line 336: When referring to the IS/Tn names, the numbers MUST be in italics.

Response: We corrected the naming format in the revised manuscript (**Lines: 305-307**). In addition, the naming of IS/Tn was corrected elsewhere in the manuscript if needed.

Re: Spectrum01635-24R1 (Resistome phylogenetics of multidrug-resistant *Shigella* isolated from diarrheal patients)

Dear Dr. Zahirul Islam:

Your manuscript has been accepted, and I am forwarding it to the ASM production staff for publication. Your paper will first be checked to make sure all elements meet the technical requirements. ASM staff will contact you if anything needs to be revised before copyediting and production can begin. Otherwise, you will be notified when your proofs are ready to be viewed.

Sincerely,
Cheryl Andam
Editor
Microbiology Spectrum

Reviewer #1 (Comments for the Author):

I thank the authors for addressing all my remarks and providing more details for their study

Reviewer #2 (Comments for the Author):

All my concerns have been addressed, however I recommend using the comma "," to separate resistance genes in the text (rather than a hyphen).